# Sterilization efficacy of a warm-air circulation system in a vaporized hydrogen peroxide sterilizer

**Mi Suk Noh**[1], **Su Yeon Lee**[2], **Eun Gyu Lee**[2], **Jonghyun Shin**[3,4], **Sang-Il Lee**[2], **Bong-Hyun Jun**[3]*

**1** Bio&Medical Research Center, Bio Business Division, Korea Testing Certification Institute, Gunpo, Gyeonggi-do, Republic of Korea, **2** LOWTEM CO., LTD., Gunpo, Gyeonggi-do, Republic of Korea, **3** Department of Bioscience and Biotechnology, Konkuk University, Seoul, Republic of Korea, **4** Department of Chemical and Biomolecular Engineering, University of California, Berkeley, California, United States of America

* bjun@konkuk.ac.kr

## Abstract

### Background

Vaporized hydrogen peroxide ($VH_2O_2$) sterilization is a widely recognized method for low-temperature sterilization of medical devices. To overcome limitations associated with residual moisture and maximize sterilization efficacy, the integration of a warm-air circulation system (WACS) is proposed as a robust solution.

### Objective

This study aimed to evaluate the effects of the WACS by comparing temperature uniformity and moisture removal with and without its operation. Additionally, the study assessed whether the WACS contributes to effective sterilization under conditions involving complex medical devices.

### Methods

Temperature variations within the sterilization chamber and medical devices were monitored using sensors. The impact of the WACS on sterilization efficacy was assessed by introducing artificial moisture to the loading packages in cycles performed with and without the system. Sterilization efficacy was assessed through tests with various items, including process challenge devices (PCDs), flexible and rigid lumens, and dual flexible endoscope dummies.

### Results

The WACS maintained a stable temperature of approximately 55～60 °C and effectively removed moisture throughout sterilization cycles, ensuring successful sterilization of all items.

**Data availability statement:** All relevant data are within the manuscript and its Supporting Information files.

**Funding:** This study was supported by the Ministry of SMEs and Startups (MSS), Korea, through the Korea Technology and Information Promotion Agency for SMEs (TIPA) (Grant No. S2634352). This research was also supported by Konkuk University during a sabbatical leave.

**Competing interests:** The authors have read the journal's policy and declare the following competing interests: Some of the authors (Su Yeon Lee, Eun Gyu Lee, and Sang-Il Lee) are employees of LOWTEM Co., Ltd., which provided products and the sterilizer used in this study. This does not alter our adherence to PLOS ONE policies on sharing data and materials. The other authors have declared that no competing interests exist.

## Conclusion

This study demonstrates that the WACS significantly improves sterilization effectiveness by ensuring uniform temperature and effective moisture removal. This additional system in the $VH_2O_2$ sterilizers will enhance the safety and sterilization efficacy of complex medical devices in healthcare settings.

## Introduction

Sterilization is essential for the infection control of medical devices and is strictly regulated by organizations such as the World Health Organization (WHO) and the Centers for Disease Control and Prevention (CDC) [1,2]. Traditional sterilization relies on high-temperature autoclaving for heat-stable devices [3–5]. While ethylene oxide (EO) gas has been widely used as a low-temperature alternative, it is limited by long cycle times due to aeration and concerns regarding toxicity [3–7]. Consequently, low-temperature vaporized hydrogen peroxide ($VH_2O_2$) sterilization has gained importance due to its rapid cycle times and safety profile, as it decomposes into water and oxygen [5–8]. $VH_2O_2$ sterilization effectively eliminates microorganisms at low temperatures, protecting delicate devices while ensuring sterility [6–8].

Reusable medical devices require thorough pre-treatment, including cleaning and drying, before low-temperature sterilization to prevent residual moisture that can cause sterilization failure and device damage [9]. Current $VH_2O_2$ sterilizers use vacuum during pre-conditioning to remove moisture, but residual moisture often remains [9,10]. Insufficient heat supply during vacuum drying can lead to condensation inside the sterilization chamber, destabilizing hydrogen peroxide vapor distribution and impairing sterilization [9,11,12].

Additionally, cooled external air influx during the ventilation phase induces temperature fluctuations and cold spots, reducing sterilization uniformity [13]. Minimizing hydrogen peroxide vapor condensation and maintaining its concentration are essential for sterilization effectiveness [13,14]. To overcome these challenges, a warm-air circulation system (WACS) was developed, incorporating a fan and heater inside the chamber to distribute warm air evenly and quickly raise chamber temperature.

In this study, we propose a method for improving sterilization performance by applying a WACS to avoid reducing sterilization effects and errors during the sterilization process caused by residual moisture in a $VH_2O_2$ sterilizer. Evaluation focused on the system's ability to reduce residual moisture and temperature variation, thereby optimizing $VH_2O_2$ sterilization performance for complex medical devices.

## Materials and methods

### Ethical statement

This study evaluated the sterilization efficacy of a medical device using commercially available biological indicators and standard test soils. The study did not involve human participants, human biological material, or animal subjects. Therefore, approval from an Institutional Review Board (IRB) or ethics committee was not required.

## Test sterilizers with the WACS

We used a VH$_2$O$_2$ sterilization system (Crystal 120M sterilizer, LOWTEM, Korea) equipped with a WACS inside the sterilizer chamber (Fig 1). The WACS consisted of a fan and heater located at the rear of the chamber to raise and equalize the air temperature inside the chamber during the sterilization process. In this system, the fan rotates to force air from the front to the back of the chamber. The air then flows over a heater located directly behind the fan, raising its temperature. The heated air flows back to the front of the fan and then uniformly distributes throughout the entire chamber. This process repeats continuously, ensuring the air inside the chamber circulates and heats up consistently.

## Comparison of temperature changes with and without the WACS

**Temperature distribution at different chamber locations.** The temperature was measured at five locations in the chamber. Temperature sensors (paperless recorder, GP10, YOKOGAWA, Japan) were placed at five locations inside the chamber: front middle, middle, rear middle, top, and bottom, and the WACS was operated for 20 min (S1 Fig a, b). Under the same conditions, the temperature at each position was measured without operating WACS to compare differences in temperature rise.

**Temperature changes in the medical device.** The temperature change of medical scissors in the chamber was examined depending on the operation of the WACS for 20 min (Fig S1 Fig c). A temperature sensor was attached to the hinge of the stainless-steel scissors in the medical device loads, and the scissors with the temperature sensor were placed in the chamber. The temperature of the scissors was measured with the WACS in operation. Under the same conditions, the temperature was measured without the WACS.

## Residual moisture removal test

A total of 24 types of medical devices (S1 Table), composed of both metal and non-metal materials —including stainless steel specula, silicone tubes, PVC tubes, stainless steel scissors, and stainless steel lumens—were individually packaged in sterilization pouches (Perpak™ Sterilization Tyvek Pouch, KM, South Korea, Lot No. 201120-B1, 150 mm x 200M). Each sterilization pouch was injected with 0.5 g of distilled water using a pipette (Research plus variable 100–1000 μL, Eppendorf, Germany). The total amount of water injected into the validation loads was 12 g (S2 Fig a,b).

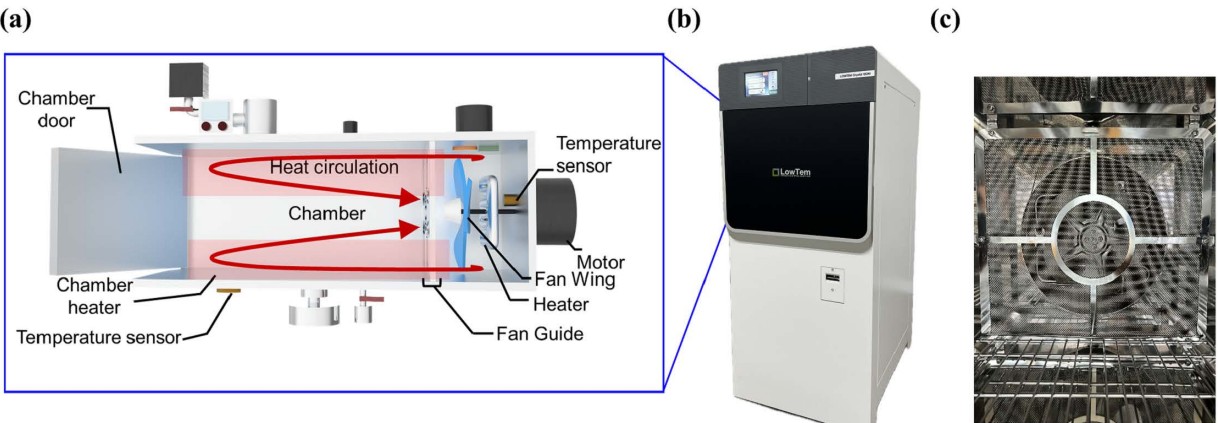

**(a)** **(b)** **(c)**

**Fig 1. Schematic diagram of WACS in the test sterilizer. (a)** The air heated by the heater is distributed throughout the chamber by convection, induced by the operation of the fan. **(b)** Appearance of the test sterilizer. **(c)** WACS mounted on the back side of the chamber.

Before sterilization, the total weights of the validation loads, including the added moisture, were measured using a precision balance (initial weight, CF-1000; AND, Japan). The validation loads were then placed on the top and bottom shelves of the sterilizer and processed with a Special Full cycle (54 min). After completing the sterilization process, the weights of the processed validation loads were measured again for comparison with their initial weights. The presence or absence of residual moisture inside each pouch was visually observed after measuring the weight.

The difference in weight before and after sterilization indicates the amount of moisture removed by the sterilizer equipped with the WACS.

### Sterilization efficacy test with the WACS

*Preparation of test organisms and carriers.* In the sterilization efficacy test, *Geobacillus stearothermophilus* spores (ATCC 7953) were used as the representative biological indicator. Selected in accordance with ISO 11138−1 [15], this organism is highly resistant to vaporized hydrogen peroxide and was inoculated with a population of $\geq 1.0 \times 10^6$ spores to validate a Sterility Assurance Level (SAL) of $10^{-6}$ as required by ISO 22441 [12]. Since *G. stearothermophilus* is identified as the most resistant microorganism to vaporized hydrogen peroxide, confirming its inactivation ensures the effective sterilization of the entire load [1,10,12]. Two types of spores were prepared: (i) glass fiber spore discs (GFTS-06, Crosstex, Englewood, Colorado), $2.7 \times 10^6$ spores per disc, and (ii) stainless steel wires (0.25 mm × 60 mm) inoculated with a spore suspension (VSG-07, Crosstex, Englewood, Colorado), $2.9 \times 10^6$ spores per wire.

*Recovery method.* To confirm the recovery rate of *G. stearothermophilus* (ATCC 7953, $2.9 \times 10^6$/20 μl), inoculated stainless-steel wires (0.25 mm × 60 mm) were used. The procedure followed the guidelines specified in ISO 11737−1:2018, Annex C, Validation of Bioburden Recovery Efficiency [16]. The stainless-steel wire was inoculated with $2.53 \times 10^6$ CFU/unit and placed into the eluate. The eluate containing the inoculated wire was vortexed for 5 min. The vortexed sample was transferred to TSA media (Becton, Dickinson and Company, USA) and incubated at 55~60 °C for 24 hours to observe spore growth (Incubator, DS-DI 150, Dongseo Science Co., Ltd). Three trials were conducted, and the average recovery was calculated.

### Sterilization efficacy tests for flexible PCDs and flexible lumens with and without the WACS under residual moisture conditions

*Sterilization efficacy under residual moisture conditions.* Each item to be sterilized was individually packaged in a sterilization pouch, and water (0.5 g) was injected into the sterilization pouch containing the item. The prepared items were processed with test flexible PCDs (Process Challenge Device, one-sided open type, 2 mm inner diameter × 1,500 mm in length) and flexible lumens (straight type, 1 mm inner diameter × 2,000 mm in length) in Fig 3a.

The specific experimental conditions for the sterilization cycles with and without the WACS were established as follows:

- With WACS: a biological indicator (BI) disc (glass fiber disc, inoculated with $2.7 \times 10^6$/disc, GKE, Germany) was inserted into the head of the test flexible PCD. Ten flexible PCDs, each containing a BI disc, were placed alongside the moisture-loaded items in validation trays to be exposed to a half-cycle of the $VH_2O_2$ sterilization process. After completing the cycle, the BI discs were removed from the test flexible PCDs, transferred to TSB media (Becton, Dickinson and Company, USA), and incubated at 55~60 °C for 7 days to check for bacterial growth. Similarly, stainless steel wires inoculated with $2.9 \times 10^6$ spores were inserted into flexible lumens and subjected to sterilization with moisture-loaded items.

- Without WACS: a BI disc (glass fiber disc, inoculated with $1.0 \times 10^6$/disc, GKE, Germany) was inserted into the head of the test flexible PCD. Ten flexible PCDs, each containing a BI disc, were placed alongside the moisture-loaded items in validation trays to be exposed to a half-cycle of the $VH_2O_2$ sterilization process. After completing the cycle, the BI discs were removed from the test flexible PCDs, transferred to TSB media (Becton, Dickinson and Company, USA), and incubated at 55~60 °C for 7 days to check for bacterial growth. Similarly, stainless steel wires inoculated with $2.2 \times 10^6$

spores/10 µL (True indicating, USA) were inserted into flexible lumens and subjected to sterilization with moisture-loaded items.

**Rigid lumen sterilization test.** Ten sets of narrow stainless steel lumens—one with an inner diameter of 1 mm and a length of 500 mm, and the other with 0.7 mm inner diameter and 500 mm—were prepared (Fig 3b). A wire BI carrier (0.25 mm outer diameter × 60 mm length, inoculated with $2.9 \times 10^6$ spores) was inserted into each stainless-steel lumen, and the lumens were individually packaged in sterilization pouches. The prepared lumens were then divided into two validation trays, with five lumens per tray. These validation trays were subsequently loaded into the sterilizer for a half-cycle sterilization process. After completing the process, the BI wire carriers were removed from the test lumens and transferred to tubes containing 10 ml of TSB medium. The media tubes were then incubated at 55~60 °C for 7 days to observe microbial growth.

**Process challenge device (PCD) test under worst-case conditions.** Ten sets of one-sided open PCDs—one with a 1 mm inner diameter and 1,500 mm length, and the other with a 2.0 mm inner diameter and 1,500 mm length—were prepared. Each PCD contained a glass fiber BI disc inoculated with $2.7 \times 10^6$ *G. stearothermophilus* spores. The PCDs were loaded into the sterilizer chamber along with six sets of challenge packs (Micro Instrument Case, Sungkwang Meditech Co., Ltd., South Korea), which included two silicone mats to simulate worst-case conditions (Fig 3c). After completing the half-cycle sterilization process, the spore discs inserted inside the test PCDs were transferred to tubes containing 10 ml of TSB. The media tubes were then incubated at 55~60 °C for 7 days to observe microbial growth.

**Dual-channel flexible endoscope sterilization test.** The sterilization efficacy on a dual-channel flexible endoscope using an endoscope surrogate device ("SPO-PRO" Version 2.0, Spypach, Austria) was validated. The dual-channel surrogate device had one with a 1.0 mm inner diameter × 990 mm in length, and the other with a 1.0 mm inner diameter × 850 mm in length. Three glass fiber BI discs, each inoculated with $2.7 \times 10^6$ *G. stearothermophilus* spores, were inserted into the BI chamber of the surrogate device (Fig 3d). The device was then placed in a sterilization tray and double-wrapped with a sterilization wrap. The tray was positioned at the bottom of the sterilization chamber, and a half-cycle sterilization process was performed (Fig 3e). After the process was completed, the spore discs were removed and transferred to TSB media. The media tubes were incubated at 55~60 °C for 7 days to observe microbial growth.

**Statistical analysis.** All measurements were independently repeated three times. Quantitative results were summarized as mean ± standard deviation (*SD*), and statistical differences between conditions were evaluated using an independent Student's *t*-test. A *p*-value of < 0.05 was considered statistically significant, and highly significant differences ($p < 0.001$) were noted where applicable. All statistical analyses were performed using Microsoft Excel (Microsoft Corporation, Redmond, WA, USA).

## Results

### Comparison of temperature variation with and without the WACS

***Temperature changes at different chamber locations.*** With the WACS (Fig 2a), the temperature was more consistent across different locations, with the maximum temperature difference of 8.3 °C at the start point and as low as 0 °C after 20 minutes. In addition, as time progressed, the temperature difference between the top and bottom locations decreased significantly, indicating the effective circulation of warm air (S2 Table). The temperatures measured at all five locations inside the chamber for 30 minutes increased uniformly, with no difference between locations. Without the WACS (Fig 2b), however, the temperature difference between locations is much larger, starting at 19.4 °C and gradually decreasing to 11.1 °C after 20 minutes (S2 Table). This indicated less uniform heating and a higher variation in temperature across different positions inside the chamber.

The results demonstrated that the WACS facilitated uniform temperature distribution across all positions within the chamber.

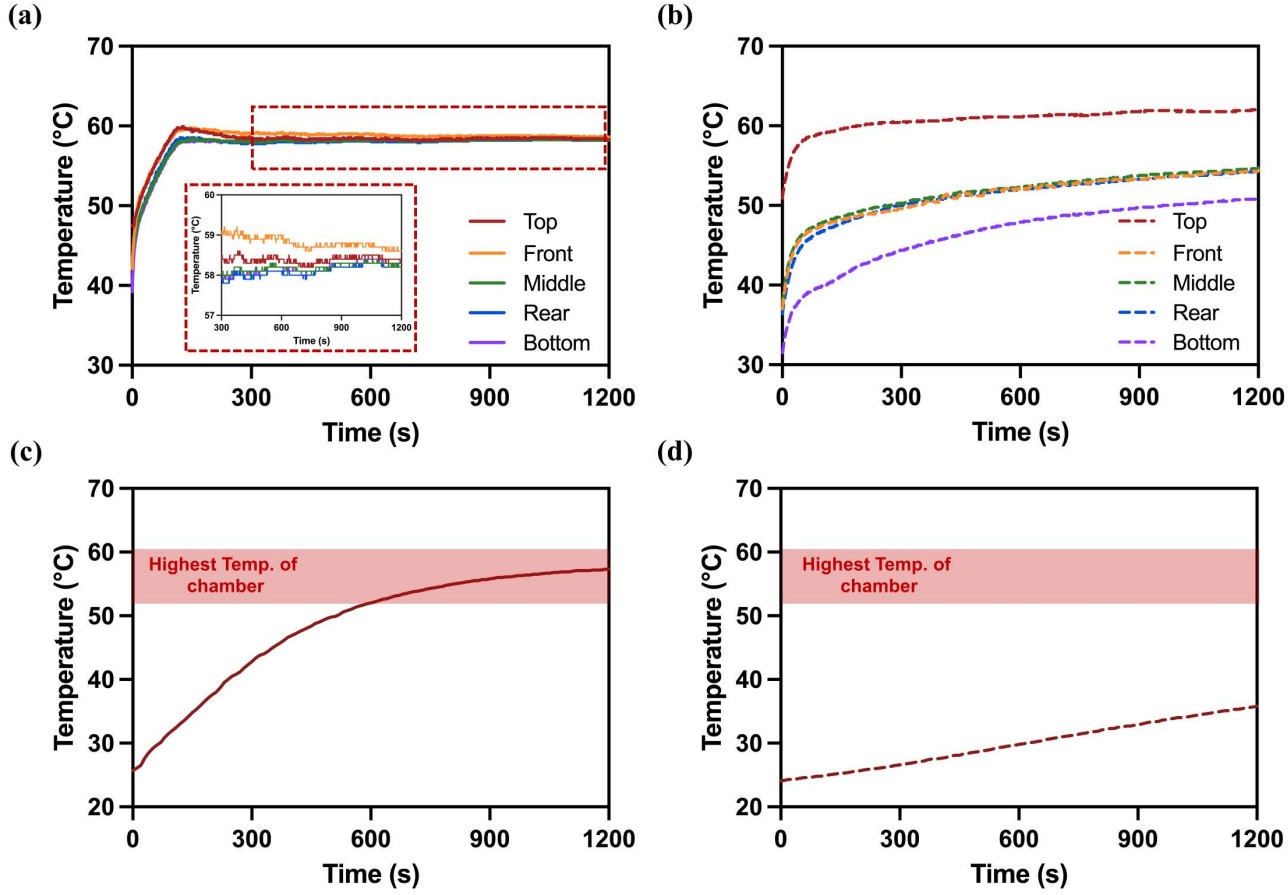

**Fig 2. Temperature changes by locations and the hinge of scissors loaded along with medical devices in the chamber.** Temperature changes by locations in the chamber with (a) or without operation of the WACS (b) were measured. Temperature changes of the hinge of scissors loaded along with medical devices inside the chamber were measured (c) with or (d) without operation of the WACS.

**Temperature of the medical device.** With the WACS (Fig 2c), the surface temperature of the scissors increased within the first 5–10 minutes, rising to 52.0 °C at 10 minutes. After 20 minutes, the temperature reached 57.3 °C, closely matching the chamber temperature. This indicated the efficiency of WACS for the loaded items (S3 Table).

Without the WACS (Fig 2d), however, the temperature increase was significantly slower, reaching only 35.8 °C after 20 minutes, which is 21.5 °C lower than with the WACS. Over 20 minutes of performance, the total temperature increase was just 11.7 °C, reflecting the less effective heating of the device without the WACS (S3 Table).

A higher temperature increase in the items to be sterilized can more effectively remove the remaining moisture. Thus, the WACS, which increases the temperature rapidly for the initial 5 minutes, provides a favorable thermal environment that may help mitigate the potential condensation of hydrogen peroxide gas.

## Residual moisture removal efficacy

Residual moisture removal efficacy was confirmed by measuring and comparing the total weight of validation loads before and after sterilization. The validation loads consisted of 24 metal and non-metal medical devices (S1 Table), each packaged in a pouch containing 0.5 g of distilled water (total 12 g of moisture) before and after the sterilizer by Special Full cycle with the WACS.

As shown in Table 1, the weights of the loads were measured before and after the sterilization cycles. The system with WACS achieved a mean moisture reduction of 12.92 ± 0.43 g, which was significantly higher than the 8.10 ± 0.13 g reduction observed without WACS ($p < 0.001$). The total weight reduction of 12.92 g slightly exceeded the 12 g of intentionally injected water, which is likely attributable to the evaporation of intrinsic residual moisture initially present within the complex internal structures of the validation loads.

In addition, the visual inspection of the medical devices inside the pouch after each sterilization cycle revealed no residual moisture or condensation. This observation was corroborated by the weight reduction data, confirming the complete removal of the added moisture.

The WACS completely removed added moisture from the pouched loads during sterilization cycles, leaving no visible residue. This demonstrates that the system effectively eliminates moisture from sterilized items and prevents issues like incomplete drying, condensation, and potential sterilization failure.

## Sterilization efficacy tests for flexible PCDs and flexible lumens with and without the WACS under residual moisture conditions

***Sterilization efficacy for items containing residual moisture.*** The sterilization efficacy was verified with *G. stearothermophilus* spores under the sterilization half-cycle with added moisture conditions, using two types of sterilization evaluation accessories: (a) flexible PCDs and (b) flexible lumens (Fig 3a). The numbers of surviving test organisms after sterilization with and without WACS are shown in Table 2. The bioburden recovery efficiency of *G. stearothermophilus* spores (ATCC 7953, $2.9 \times 10^6/20$ ml) was measured at about 95% in three reproductive tests.

For sterilization with WACS, (i) glass-fiber BI discs inoculated with *G. stearothermophilus* spores, inserted into 10 flexible PCDs of 2 mm × 1,500 mm, loaded into two sterilization trays containing 11 kg of medical devices with added moisture, all showed negative results for growth after undergoing sterilization half-cycles. No bacteria were observed in the negative control group without the inoculated BI discs, whereas bacteria were observed in the positive control group with non-sterilized BI discs. (ii) The stainless-steel wires inoculated with *G. stearothermophilus* spores were inserted into 10 flexible lumens of 1 mm × 2,000 mm loaded in two sterilization trays, including 11 kg of medical devices with added moisture, and were all found to have negative growth results after sterilization half-cycles. No bacteria were observed in the negative control group with un-inoculated wires. In contrast, bacteria were observed in the positive control group with non-sterilized wires after inoculation.

For sterilization without WACS, (i) glass-fiber BI discs inoculated with *G. stearothermophilus* spores, inserted into 10 flexible PCDs of 2 mm × 1,500 mm, loaded into two sterilization trays containing 11 kg of medical devices with added moisture, all showed positive results for growth after undergoing sterilization half-cycles. No bacteria were observed in the negative control group without the inoculated BI discs, whereas bacteria were observed in the positive control group with non-sterilized BI discs. (ii) The stainless-steel wires inoculated with *G. stearothermophilus* spores were inserted into 10 flexible lumens of 1 mm × 2,000 mm loaded in two sterilization trays, including 11 kg of medical devices with added moisture, and were found to have positive growth in 2–4 out of 10 replicates after sterilization half-cycles. No bacteria were

**Table 1. Moisture reduction before and after sterilization process with and without a WACS.**

| WACS Condition | Initial Weight (g) | Weight after Sterilization (g) | Moisture Reduction (g) |
|---|---|---|---|
| With | 2006.74 ± 3.46 | 1993.82 ± 3.51 | 12.92 ± 0.43*** |
| Without | 2004.42 ± 4.75 | 1996.32 ± 4.62 | 8.10 ± 0.13 |

Data are presented as mean ± standard deviation (*SD*) of three independent replicates (*n* = 3). *** Indicates a significant difference compared to the 'Without WACS' group ($p < 0.001$, Student's *t*-test).

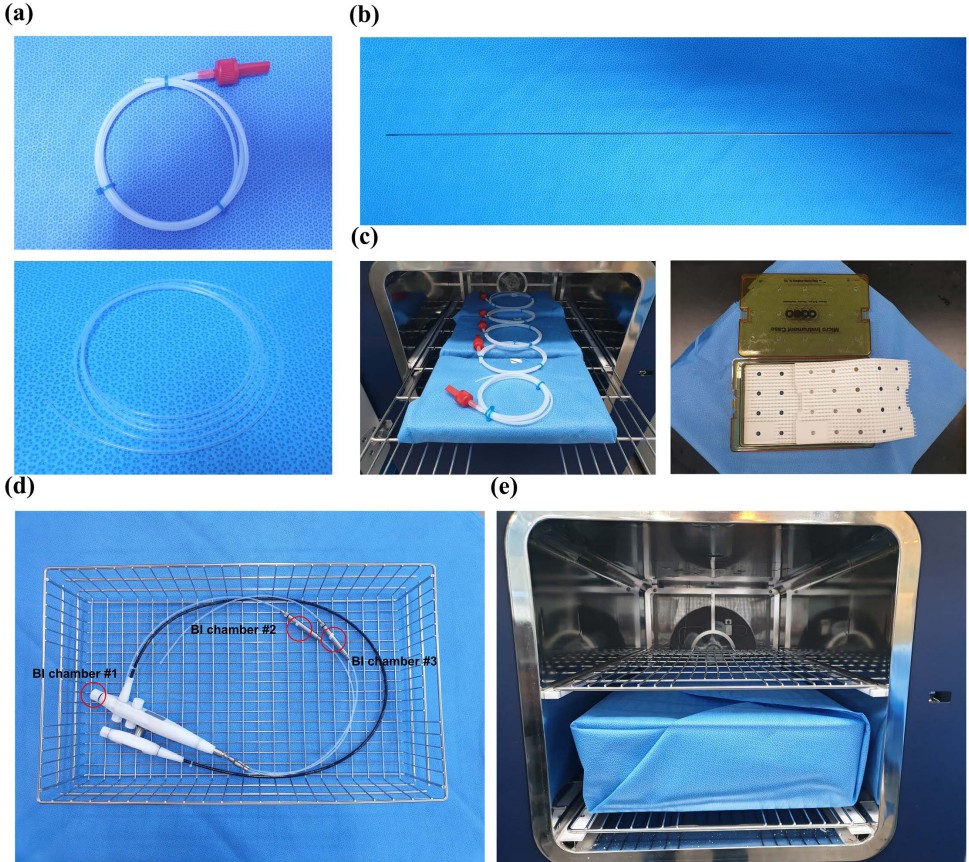

**Fig 3. Diverse PCDs, flexible endoscope with inserted BIs for sterilization efficacy tests. (a)** Two types of items with residual moisture: flexible PCD (upper: one side open type, 2 mm inner tube × 1,500 mm in length) and flexible lumens (lower: straight type, 1 mm inner tube × 2,000 mm in length); **(b)** Rigid lumen (1 mm inner tube × 500 mm in length); **(c)** PCDs with challenge packs (right side) for sterilization efficacy tests; **(d)** Dual-channel flexible endoscope with inserted BI discs placed inside the chamber **(e)**.

**Table 2. Comparison of sterilization efficacy for flexible PCDs and lumens with and without the WACS under residual moisture conditions.**

| WACS | Repeat | A | B | PC | NC |
|---|---|---|---|---|---|
| **With** (A: Flexible PCDs, B: Flexible lumens) | 1 | 0/10 | 0/10 | 1/1 | 0/1 |
| | 2 | 0/10 | 0/10 | 1/1 | 0/1 |
| | 3 | 0/10 | 0/10 | 1/1 | 0/1 |
| **Without** (A: Flexible PCDs, B: Flexible lumens) | 1 | 10/10 | 4/10 | 1/1 | 0/1 |
| | 2 | 10/10 | 2/10 | 1/1 | 0/1 |
| | 3 | 10/10 | 2/10 | 1/1 | 0/1 |

Data are presented as (No. of positive cultures) / (Total no. of samples). A positive result indicates bacterial growth and signifies a non-sterile condition. Abbreviations: PC, positive control; NC, negative control.

observed in the negative control group with un-inoculated wires. In contrast, bacteria were observed in the positive control group with non-sterilized wires after inoculation.

Specifically, in the half-cycles without the WACS, the flexible PCDs exhibited a 100 % failure rate, with bacterial growth observed in all samples (30/30) across three independent trials. The flexible lumens showed inconsistent efficacy, with

growth detected in 2–4 samples out of 10 in each trial (8/30 growth). In contrast, the half-cycles with the WACS successfully achieved complete inactivation (0/30 growth) for both flexible PCDs and flexible lumens, demonstrating consistent sterilization efficacy regardless of the device type.

## Sterilization efficacy tests for various PCDs and flexible endoscopes with the WACS

Tests were conducted to validate the sterilization efficacy of the WACS-integrated system against highly challenging medical devices, including complex flexible endoscopes and narrow lumens, which represent the critical targets for advanced $VH_2O_2$ sterilization.

***Rigid lumen test***. The sterilization efficacy of rigid lumens (Fig 3b) validated with *G. stearothermophilus* spores under the sterilizer by half-cycle conditions is shown in Table 3. The stainless-steel wires inoculated with *G. stearothermophilus spores*, inserted into 10 each of the 1 mm × 500 mm and 0.7 mm × 500 mm stainless steel rigid lumens loaded in two sterilization trays, were found to have negative results for growth when incubated for 7 days at 55~60 °C after the sterilization half-cycles. Under the same incubation conditions, no bacteria were observed in the negative control group with un-inoculated wires. In contrast, bacteria were observed in the positive control group with non-sterilized wires after inoculation.

**PCD testing under worst-case conditions.** The sterilization efficacy of the PCDs (Fig 3c) under the worst-case condition, validated with *G. stearothermophilus* spores under the sterilizer by half-cycle condition, is shown in Table 3. The glass-fiber BI discs, inoculated with *G. stearothermophilus* spores, were inserted into 10 each of the 1 mm × 1,500 mm and 2 mm × 1,500 mm flexible PCDs, loaded along with six sets of challenge packs containing two silicone mats each, and were found to be nonviable during 7-day incubation period at 55~60 °C after undergoing sterilization half-cycles. Under the same incubation conditions, no bacteria were observed in the negative control group without inoculated BI discs, whereas bacteria were observed in the positive control group with non-sterilized BI discs.

**Dual-channel flexible endoscope sterilization test.** The sterilization efficacy of the dual-channel flexible endoscope (Fig 3d, 3e) using an endoscope surrogate device validated with *G. stearothermophilus* spores under the sterilizer by half-cycle conditions is shown in Table 4. Three glass-fiber BI discs, inoculated with *G. stearothermophilus* spores, inserted into the BI chamber of the surrogate device loaded in a sterilization tray, and double-wrapped with sterilization wraps, showed no viability during the 7-day incubation period at 55~60 °C after the sterilization half-cycles. Under the same incubation conditions, no bacteria were observed in the negative control group without inoculated BI discs, whereas bacteria were observed in the positive control group with non-sterilized BI discs.

**Table 3. Sterilization efficacy results for rigid lumens and flexible PCDs.**

| WACS | Repeat | A | B | PC | NC |
|---|---|---|---|---|---|
| **Rigid Lumens** (A: 1 mm x 500 mm B: 0.7 mm x 500 mm) | 1 | 0/10 | 0/10 | 1/1 | 0/1 |
| | 2 | 0/10 | 0/10 | 1/1 | 0/1 |
| | 3 | 0/10 | 0/10 | 1/1 | 0/1 |
| **Flexible PCDs** (A: 1 mm x 1,500 mm B: 2 mm x 1,500 mm) | 1 | 0/10 | 0/10 | 1/1 | 0/1 |
| | 2 | 0/10 | 0/10 | 1/1 | 0/1 |
| | 3 | 0/10 | 0/10 | 1/1 | 0/1 |

Data are presented as (No. of positive cultures) / (Total no. of samples). A positive result indicates bacterial growth and signifies a non-sterile condition. Abbreviations: PC, positive control; NC, negative control.

**Table 4. Sterilization efficacy results for dual-channel flexible endoscopes.**

| Types | Repeat | A | B | C | PC | NC |
|---|---|---|---|---|---|---|
| Dual-channel flexible endoscopes (A: BI chamber #1, B: BI chamber #2, C: BI chamber #3) | 1 | 0/1 | 0/1 | 0/1 | 1/1 | 0/1 |
| | 2 | 0/1 | 0/1 | 0/1 | 1/1 | 0/1 |
| | 3 | 0/1 | 0/1 | 0/1 | 1/1 | 0/1 |

Data are presented as (No. of positive cultures) / (Total no. of samples). A positive result indicates bacterial growth and signifies a non-sterile condition.
Abbreviations: PC, positive control; NC, negative control.

## Discussion

Sterilization is essential for the safety of medical devices, with single-use medical devices and reusable devices requiring validated methods per manufacturer guidelines [17–22]. Due to the carcinogenicity of ethylene oxide (EO), vaporized hydrogen peroxide ($VH_2O_2$) sterilization is preferred for delicate instruments [3,4,6,7]. However, as $VH_2O_2$ systems are highly sensitive to humidity, which can cause vapor condensation and compromise sterilization, thorough humidity removal is vital for efficacy [10,23–25]]. Therefore, this study assessed the effect of introducing a warm-air circulation system (WACS) to promote moisture removal and optimize sterilization efficiency in $VH_2O_2$ processes.

The WACS in $VH_2O_2$ processes significantly improved temperature uniformity inside the sterilization chamber, eliminating cold spots that impair sterilization, especially in complex device structures. Specifically, by rapidly increasing the temperature during the initial phase (e.g., the first 5 minutes), the WACS provides a favorable thermal environment that effectively mitigates the potential condensation of hydrogen peroxide gas. Moisture removal was enhanced, as demonstrated by weight loss measurements, confirming effective mitigation of residual moisture. Notably, the comparative tests under residual moisture conditions highlighted the differing levels of challenge presented by the devices. The flexible PCDs resulted in total sterilization failure (100% growth) without the WACS, whereas the flexible lumens showed partial and inconsistent failures (20–40% growth). This confirms that the fabricated flexible PCDs effectively simulate a worst-case scenario that is more rigorous than standard lumens. Most importantly, the WACS successfully overcame these challenging conditions, ensuring complete sterility for both the highly resistant PCDs and the variable lumens, thereby eliminating the risk of inconsistent sterilization outcomes caused by residual moisture.

This elimination of inconsistency has profound clinical implications. Previous studies have emphasized that residual moisture in complex channels can freeze during the vacuum phase, blocking narrow lumens and hindering the efficient contact of the sterilant with microorganisms [9]. Consequently, this mechanism directly leads to load non-sterility. By actively removing this residual moisture, the WACS provides a critical 'safety margin,' preventing issues such as condensation and potential sterilization failures associated with incomplete drying.

Various devices, including complex lumens and endoscopes, achieved complete sterilization even under challenging conditions verified by BIs. Recent reviews indicate that ensuring sterilant penetration into narrow lumens remains a significant challenge for conventional systems, as residual moisture can physically obstruct these pathways [9]. Our results demonstrate that the WACS facilitates effective vapor penetration even in these difficult-to-sterilize geometries, expanding the applicability of $VH_2O_2$ sterilization to more complex and challenging medical devices.

Finally, regarding material safety, the WACS ensures stability by maintaining the temperature below 60°C. This aligns with the ISO 22441 standard for low-temperature sterilization [12]. As supported by comprehensive reviews, this low-temperature profile is fully compatible with heat-sensitive medical polymers and adhesives used in flexible endoscopes [10]. Thus, the system ensures material compatibility and device stability while delivering robust sterilization efficacy for complex geometries that are typically difficult to sterilize with conventional $VH_2O_2$ systems.

## Conclusion

This study demonstrates that WACS significantly enhances the reliability and effectiveness of $VH_2O_2$ sterilization by addressing key challenges related to moisture and temperature variability. Notably, the process was confirmed to achieve a Sterility Assurance Level (SAL) of $10^{-6}$ even under defined challenge conditions. Consequently, this technology shows great potential for improving the safety and consistency of sterilizing complex medical devices and for supporting effective reprocessing practices in healthcare settings.

## Supporting information

**S1 Fig. Location of temperature sensors in the chamber and on medical devices.** (a) Top view of the chamber (sensor positions: a. front, b. middle, c. rear). (b) Front view of the chamber (sensor positions: d. top, e. bottom). (c) A temperature sensor attached to the hinge surface of surgical scissors within the medical device load.
(TIF)

**S2 Fig. Experimental setup for the residual moisture removal test.** (a) Preparation of representative medical device packages injected with water to simulate wet conditions (see S1 Table for the full list of 24 devices). (b) Configuration of the medical device load inside the sterilization chamber. This setup was used to visually verify moisture reduction efficiency with and without the Warm-Air Circulation System (WACS).
(TIF)

**S1 Table. List of 24 types of medical devices used in the residual moisture removal test and the sterilization efficacy test.** Details include the device name, manufacturer, and material composition.
(DOCX)

**S2 Table. Comparison of continuous temperature changes by location in the chamber.** Data confirms temperature profiles with and without the Warm-Air Circulation System (WACS).
(DOCX)

**S3 Table. Comparison of continuous temperature changes at the scissor hinge.** Data represents temperature monitoring at the hinge of scissors loaded inside the chamber, with and without the WACS.
(DOCX)

## Author contributions

**Conceptualization:** Mi Suk Noh, Bong-Hyun Jun.

**Data curation:** Mi Suk Noh, Su Yeon Lee, Eun Gyu Lee, Jonghyun Shin, Sang-Il Lee.

**Formal analysis:** Su Yeon Lee, Eun Gyu Lee, Sang-Il Lee.

**Methodology:** Mi Suk Noh.

**Resources:** Mi Suk Noh.

**Supervision:** Mi Suk Noh, Jonghyun Shin, Bong-Hyun Jun.

**Validation:** Mi Suk Noh, Su Yeon Lee, Eun Gyu Lee, Sang-Il Lee.

**Visualization:** Mi Suk Noh, Su Yeon Lee, Jonghyun Shin, Bong-Hyun Jun.

**Writing – original draft:** Mi Suk Noh, Bong-Hyun Jun.

**Writing – review & editing:** Mi Suk Noh, Bong-Hyun Jun.

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
