## [Decision Letter · Decision Letter 0]

17 Nov 2025

PONE-D-25-51990Sterilization efficacy of a warm-air circulation system in a vaporized hydrogen peroxide sterilizerPLOS ONE

Dear Dr. Jun,

Thank you for submitting your manuscript to PLOS ONE. After careful consideration, we feel that it has merit but does not fully meet PLOS ONE’s publication criteria as it currently stands. Therefore, we invite you to submit a revised version of the manuscript that addresses the points raised during the review process.

We look forward to receiving your revised manuscript.

Kind regards,

Shengqian Sun

Academic Editor

PLOS ONE

Journal Requirements:

“This study was supported by the Ministry of SMEs and Startups (Grant No. S2634352), South Korea.”

“This study was supported by the Ministry of SMEs and Startups (Grant No. S2634352), South Korea.”

“This study utilized test results provided by the Korea Testing Certification Institute (KTC, South Korea). Products and the sterilizer were provided by LowTem Co., Ltd. (South Korea). This study was supported by the Ministry of SMEs and Startups (Grant No. S2634352), South Korea. The funders had no role in study design, data collection and analysis, decision to publish, or preparation of the manuscript.”

“This study was supported by the Ministry of SMEs and Startups (Grant No. S2634352), South Korea.”

“The authors have read the journal’s policy and declare the following competing interests:

Some of the authors (Su Yeon Lee, Eun Gyu Lee, and Sang-Il Lee) are employees of LOWTEM Co., Ltd., which provided products and the sterilizer used in this study. This does not alter our adherence to PLOS ONE policies on sharing data and materials. The other authors have declared that no competing interests exist.”

We note that one or more of the authors are employed by a commercial company:  LOWTEM Co., Ltd.,

7. We note that Figure 1in your submission contain copyrighted images. All PLOS content is published under the Creative Commons Attribution License (CC BY 4.0), which means that the manuscript, images, and Supporting Information files will be freely available online, and any third party is permitted to access, download, copy, distribute, and use these materials in any way, even commercially, with proper attribution. For more information, see our copyright guidelines: http://journals.plos.org/plosone/s/licenses-and-copyright.

a. You may seek permission from the original copyright holder of Figures 1 to publish the content specifically under the CC BY 4.0 license.

Reviewers' comments:

Reviewer's Responses to Questions

**Comments to the Author**

1. Is the manuscript technically sound, and do the data support the conclusions?

Reviewer #1: Partly

Reviewer #2: Partly

2. Has the statistical analysis been performed appropriately and rigorously? 

Reviewer #1: N/A

Reviewer #2: No

3. Have the authors made all data underlying the findings in their manuscript fully available?

Reviewer #1: No

Reviewer #2: No

4. Is the manuscript presented in an intelligible fashion and written in standard English?

Reviewer #1: Yes

Reviewer #2: Yes

5. Review Comments to the Author

Reviewer #1: Dear authors,

thank you very much for giving me the opportunity to read your manuscript on the implementation of a warm-air circulation system for vapourised H₂O₂ sterilisation. It is a well-structured and extremely well-presented study. I do, however, have two major points of critique which I believe require addressing before publication. I also have a few smaller suggestions that I believe would improve the manuscript.

Major points:

- In my view, the efficacy tests (Figs 3 and 4) should include a control with the WACS turned off. For a device used in clinical practice, the expected result would be the inactivation of all test organisms, but without the tests, this cannot be ascertained. This could be an opportunity to demonstrate that efficacy is improved with the WACS.

- Currently, the manuscript lacks a discussion in the classical sense. You rightly state that effective sterilisation is important and that WACS seems beneficial. But what are the overarching implications? How often are medical devices currently insufficiently sterilised, and how would implementing a WACS change that? Do you expect the rate of healthcare-associated infections to decrease? Using a WACS seems to lead to higher temperatures of the medical devices – could this cause stability issues? Could one reduce the overall temperature and save energie or the sterilization time to save time?

Minor Points:

Introduction

- You introduced the abbreviation WACS in the abstract, please introduce it once again in the introduction.

Methods

- Please provide a list of the 24 types of medical device used in the supplements. Ideally, include the manufacturer and material information to help identify which devices are compatible with this setup, especially given the increased temperature of the devices.

- Just to make sure it is not a typo - for G. stearothermophilus you used 59°C and for the biological indicators 58°C?

Supporting Information

- Table S2 is missing.

Interesting (but by no means necessary):

- With the overall benefits of the WACS, it would be interesting to test reduced cycle times with regard to sterilisation efficacy.

Reviewer #2: The experimental design has major defaults; it must be improved.

Line numbers should be added.

"Preparation of test organisms and carriers

Two types of spores were prepared: (i) glass fiber spore discs (GFTS-06, Crosstex, Englewood, Colorado), 2.7×106 spores per disc, and (ii) stainless steel wires (0.25 mm×60 mm) inoculated with a spore suspension (VSG-07, Crosstex, Englewood, Colorado), 2.9×106 spores per wire."

Any reference for the spore load applied for the sterilization efficacy test? Also, the spores' dose-dependent effect should be tested; at least two or three spore loads should be implemented for the test.

Table 1. There is no need to show each replicate; please present the data in the style of average ± standard deviation. Also, significant differences, a t-test, or LSD should be added for the data comparison. The method should also include statistical analysis.

Fig.3. It is very confusing for the y-axis; the % growth, actual microorganism load should be presented, instead of %. Also, it is not reasonable that all of the growth for the + control shows 100% growth. Dose-dependent effect should be analyzed.

6. PLOS authors have the option to publish the peer review history of their article (what does this mean?). If published, this will include your full peer review and any attached files.

Reviewer #1: No

Reviewer #2: No

---

## [Author Response · Author response to Decision Letter 1]

28 Jan 2026

We have provided a detailed, point-by-point response to all reviewer and editor comments in the attached “Response to Reviewers” document.

All suggested revisions have been carefully addressed, and corresponding changes have been incorporated into the revised manuscript. Changes in the manuscript are shown using track changes for clarity.

We thank the Editor and Reviewers for their constructive comments, which have helped to improve the quality of our manuscript.

---

## [Decision Letter · Decision Letter 1]

25 Mar 2026

PONE-D-25-51990R1Sterilization efficacy of a warm-air circulation system in a vaporized hydrogen peroxide sterilizerPLOS One

Dear Dr. Jun,

Thank you for submitting your manuscript to PLOS ONE. After careful consideration, we feel that it has merit but does not fully meet PLOS ONE’s publication criteria as it currently stands. Therefore, we invite you to submit a revised version of the manuscript that addresses the points raised during the review process.

We look forward to receiving your revised manuscript.

Kind regards,

Shengqian Sun

Academic Editor

PLOS One

**Journal Requirements:**

Reviewers' comments:

Reviewer's Responses to Questions

**Comments to the Author**

1. If the authors have adequately addressed your comments raised in a previous round of review and you feel that this manuscript is now acceptable for publication, you may indicate that here to bypass the “Comments to the Author” section, enter your conflict of interest statement in the “Confidential to Editor” section, and submit your "Accept" recommendation.

Reviewer #2: All comments have been addressed

Reviewer #3: (No Response)

2. Is the manuscript technically sound, and do the data support the conclusions?

Reviewer #2: Yes

Reviewer #3: Yes

3. Has the statistical analysis been performed appropriately and rigorously? 

Reviewer #2: Yes

Reviewer #3: Yes

4. Have the authors made all data underlying the findings in their manuscript fully available?

Reviewer #2: Yes

Reviewer #3: Yes

5. Is the manuscript presented in an intelligible fashion and written in standard English?

Reviewer #2: Yes

Reviewer #3: Yes

6. Review Comments to the Author

**Reviewer #2:** The manuscript has been greatly revised. And I have no more questions. Thanks for the authors revising. Cheers,

**Reviewer #3:** Line 175: Please check the brackets.

Lines 228-230: Here a conclusion is already drawn. The statement about “consistent sterilisation performance“ cannot be drawn from the so presented data alone. The sentence should be removed.

Line 240: “A higher temperature increases in the items [..] can [..] remove the [..] moisture.“ Please check the grammar of this sentence. I guess the authors meant “A higher temperature increase…“.

Line 242: The so far presented data does not allow a conclusion about the removal of sterilisation inhibitors. The data only allows statements regarding the heating and temperature uniformity.

Lines 252-255: Please check the grammar of this paragraph/sentence and split the sentence into two or more shorter sentences.

Table 1: With the WACS, 12.92 g of water were removed, but only 12 g of water were added before the WACS-sterilisation. How do the authors explain the additions removal of 0.92 g?

Lines 264-266: “This demonstrates that the system effectively eliminates moisture from sterilized items and prevents issues like incomplete drying, condensation, and potential sterilization failure.“ This statement is a conclusion and should not be made in the Results part of the manuscript.

Line 301: “after sterilisation half-cycle“ is written twice in the sentence

Line 363: There is a full stop set wrongly in the figure text.

7. PLOS authors have the option to publish the peer review history of their article (what does this mean?). If published, this will include your full peer review and any attached files.

Reviewer #2: No

Reviewer #3: No

---

## [Author Response · Author response to Decision Letter 2]

1 Apr 2026

We sincerely thank you and the reviewers for your valuable comments and constructive feedback on our manuscript (PONE-D-25-51990R1).

We have carefully revised the manuscript according to all comments provided. A detailed, point-by-point response to the reviewers’ comments has been included in the “Response to Reviewers” file. In addition, a marked-up version of the manuscript with tracked changes and a clean revised version have been uploaded.

We believe that the manuscript has been significantly improved and is now suitable for publication in PLOS ONE.

Sincerely,

Bong-Hyun Jun

---

## [Decision Letter · Decision Letter 2]

6 Apr 2026

Sterilization efficacy of a warm-air circulation system in a vaporized hydrogen peroxide sterilizer

PONE-D-25-51990R2

Dear Dr. Jun,

We’re pleased to inform you that your manuscript has been judged scientifically suitable for publication and will be formally accepted for publication once it meets all outstanding technical requirements.

Kind regards,

Shengqian Sun

Academic Editor

PLOS One

Additional Editor Comments (optional):

Reviewers' comments:

Reviewer's Responses to Questions

**Comments to the Author**

1. If the authors have adequately addressed your comments raised in a previous round of review and you feel that this manuscript is now acceptable for publication, you may indicate that here to bypass the “Comments to the Author” section, enter your conflict of interest statement in the “Confidential to Editor” section, and submit your "Accept" recommendation.

Reviewer #3: All comments have been addressed

2. Is the manuscript technically sound, and do the data support the conclusions?

Reviewer #3: (No Response)

3. Has the statistical analysis been performed appropriately and rigorously? 

Reviewer #3: (No Response)

4. Have the authors made all data underlying the findings in their manuscript fully available?

Reviewer #3: (No Response)

5. Is the manuscript presented in an intelligible fashion and written in standard English?

Reviewer #3: (No Response)

6. Review Comments to the Author

Reviewer #3: (No Response)

7. PLOS authors have the option to publish the peer review history of their article (what does this mean?). If published, this will include your full peer review and any attached files.

Reviewer #3: No

---

## [Editor Report · Acceptance letter]

PONE-D-25-51990R2

PLOS One

Dear Dr. Jun,

I'm pleased to inform you that your manuscript has been deemed suitable for publication in PLOS One. Congratulations! Your manuscript is now being handed over to our production team.

Kind regards,

on behalf of

Dr. Shengqian Sun

Academic Editor

PLOS One